# Evaluating the foundations that help avert antimicrobial resistance: Performance of essential water sanitation and hygiene functions in hospitals and requirements for action in Kenya

Michuki Maina[1,2]*, Olga Tosas-Auguet[3], Jacob McKnight[3], Mathias Zosi[1],
Grace Kimemia[1], Paul Mwaniki[1], Constance Schultsz[2,4], Mike English[1,3]

1 Health Services Research Group, KEMRI-Wellcome Trust Research Programme, Nairobi, Kenya,
2 Amsterdam University Medical Centres, University of Amsterdam, Amsterdam, The Netherlands, 3 Nuffield
Department of Medicine, University of Oxford—Oxford, United Kingdom, 4 Amsterdam Institute for Global
Health and Development- Amsterdam, The Netherlands

* mmaina@kemri-wellcome.org

pone.0222922

AUSTRIA

**Data Availability Statement:** All summary data
underlying the findings are freely available in the

## Abstract

### Background

Water Sanitation and Hygiene (WASH) in healthcare facilities is critical in the provision of
safe and quality care. Poor WASH increases hospital-associated infections and contributes
to the rise of antimicrobial resistance (AMR). It is therefore essential for governments and
hospital managers to know the state of WASH in these facilities to set priorities and allocate
resources.

### Methods

Using a recently developed survey tool and scoring approach, we assessed WASH across
four domains in 14 public hospitals in Kenya (65 indicators) with specific assessments of
individual wards (34 indicators). Aggregate scores were generated for whole facilities and
individual wards and used to illustrate performance variation and link findings to specific lev-
els of health system accountability. To help interpret and contextualise these scores, we
used data from key informant interviews with hospital managers and health workers.

### Results

Aggregate hospital performance ranged between 47 and 71% with five of the 14 hospitals
scoring below 60%. A total of 116 wards were assessed within these facilities. Linked to spe-
cific domains, ward scores varied within and across hospitals and ranged between 20% and
80%. At ward level, some critical indicators, which affect AMR like proper waste segregation
and hand hygiene compliance activities had pooled aggregate scores of 45 and 35%
respectively. From 31 interviews conducted, the main themes that explained this heteroge-
nous performance across facilities and wards included differences in the built environment,

manuscript and Supporting Information files. The raw data used for this manuscript are hosted in a public repository Harvard Dataverse. DOI Information: https://doi.org/10.7910/DVN/IJUWWR.

**Funding:** MM, GK, JM, MZ and OT were supported by funds through a grant from the Economic and Social Research Council ESRCS (ES/P004938/1) awarded to ME. A Senior Research Fellowship awarded to ME by The Wellcome Trust (#207522, https://doi.org/10.35802/207522) supported PM. MM received additional support from a grant to the Initiative to Develop African Research Leaders (IDeAL) through the DELTAS Africa Initiative (DEL-15-003), an independent funding scheme of the African Academy of Sciences (AAS)'s Alliance for Accelerating Excellence in Science in Africa (AESA) and supported by the New Partnership for Africa's Development Planning and Coordinating Agency (NEPAD Agency) with funding from the Wellcome Trust (#107769, https://doi.org/10.35802/107769) and the UK government.

**Competing interests:** The authors have declared that no competing interests exist.

resource availability, leadership and the degree to which local managers used innovative approaches to cope with shortages.

## Conclusion

Significant differences and challenges exist in the state of WASH within and across hospitals. Whereas the senior hospital management can make some improvements, input and support from the national and regional governments are essential to improve WASH as a basic foundation for averting nosocomial infections and the spread of AMR as part of safe, quality hospital care in Kenya.

## Introduction

Water Sanitation and Hygiene (WASH) services in healthcare facilities are integral in the provision of safe, high-quality healthcare and an essential foundation for averting the spread of antimicrobial resistance (AMR). Facilities with inadequate WASH are associated with a higher risk of hospital-associated infections and increased environmental contamination from clinical waste [1, 2] [3]. A recent global report was published by the World Health Organization's Joint Monitoring Programme for Water Supply, Sanitation and Hygiene (JMP) highlighting the current challenges with WASH. It reviewed data from 56,000 health facilities in 120 countries and revealed that a quarter of health facilities assessed lacked water from an improved source on the premises and almost half lacked hand hygiene facilities at the points where care is provided [4]. The bulk of these WASH challenges reported are in Asia and Sub Saharan Africa [4].

To improve the state of WASH in health facilities, the World Health Organization(WHO) and United Nations Education Fund (UNICEF) developed the Water Sanitation and Hygiene Facility Improvement Tool (WASH-FIT) [5]. The tool was based on global standards of environmental health and infection prevention and control (IPC) [6, 7]. It was designed for primary care facilities in limited-resource regions to assess the state of WASH and promote self-improvement. The process of developing the WASH-FIT was an iterative process involving different stakeholders to ensure the tool was validated before rolling out.

WASH-FIT entails a process of self-assessment that focuses on achieving minimum standards for a clean and safe environment in primary care facilities. The tool was not designed for more extensive facilities with multiple inpatient units, and it was not meant to survey and compare WASH performance across hospitals and their departments. We, therefore, modified the tool to create a WASH facility survey tool (WASH FAST) for use in surveys in larger hospitals with multiple inpatient units; collecting data both at the ward and facility level and creating levels of responsibility to improve accountability for WASH [8]. The development of the WASH-FAST tool has been described in detail elsewhere [8]. Briefly, this entailed three main steps: The first step was developing an approach to produce aggregate numeric scores, to enable comparisons and tracking of hospital performance of WASH over time. Secondly, it involved modifying the assessment of hospitals providing multi-speciality care—so that relevant indicators are assessed and scored for each ward in addition to the facility as a whole. Finally, the adapted tool identifies the actors who are accountable for the issues uncovered and so are responsible for effecting positive change in WASH. A comparison of the main differences between the WASH-FIT and WASH-FAST is illustrated in Fig 1.

We herein present the findings of a survey that investigated the provision of water, sanitation, hygiene and their management in 14 Kenyan county-level hospitals using WASH-FAST.

| Assessment Tool | Continuous facility improvement tool | Assessment of WASH at Ward or specialty level | Assigning accountability for improvement of WASH |
|---|---|---|---|
| WASH FIT | ✔ | ✖ | ✖ |
| WASH FAST | ✔ | ✔ | ✔ |

**Fig 1. Comparing the WASH-FIT and WASH-FAST tools.**

For brevity, we present a subset of the results to illustrate the application of WASH-FAST and to describe the status and consistency of WASH services within and across hospitals. Full results at ward level are provided in as a supplement (S1 Table). We draw on in-depth interviews with hospital managers and frontline healthcare workers conducted in parallel to the survey, to help interpret and contextualise WASH-FAST results. We also show how the findings on WASH performance can be linked to different levels of accountability within the health system. Finally, we identify best and worst-performing indicators across the hospitals based on the scoring system, some of which are critical to safe and quality patient care with implications for emergence and transmission of antimicrobial resistance (AMR). The quantitative and qualitative results combined provide important insights on WASH to national and regional governments and hospital managers. Findings can be used to inform prioritisation of actions and resource allocation aimed at improving patient safety and reducing hospital-associated infections and AMR.

## Methods

### Ethics statement

For this study, we made every effort to ensure the quality and integrity of the research. We sought and received informed consent in all cases where this was relevant. We respected the confidentiality and anonymity of our research respondents and checked they were willing to participate in the study voluntarily. We made every effort to anonymise quotes from the study respondents. This study received approval from the Oxford Tropical research ethics committee (OXTREC) from the university of Oxford (Ref: 525–17) and from the Kenyan Medical Research Institute (Ref: KEMRI/SERU/CGMR-C//086/3450).

### Study setting

The survey was carried out across 14 public county hospitals (formerly district hospitals) in Kenya. Hospitals are located in high and low malaria-endemic regions and represent a diverse selection with varying bed capacities both in urban and rural areas. All facilities participating in the study provide multi-speciality care, including at least maternity services and inpatient neonatal, paediatric, medical and surgical care. The hospitals are part of the Clinical Information Network (CIN) of the Kenya Ministry of Health. The CIN was set up to collate data from paediatric inpatient units to promote development and adoption of evidence-based clinical guidelines [9] and is coordinated by the Kenya Medical Research Institute (KEMRI) Wellcome Trust Research Programme.

## Survey preparation and data collection

As part of WASH-FAST, a data collection tool and instructions booklet were designed to allow the systematic and reproducible collection of data across survey hospitals. The booklet outlines the steps that are to be followed by the survey team, from the time they reach the facility, to undertake a systematic assessment of WASH in all hospital areas (S1 File). The WASH-FAST booklet contains 34 indicators for each hospital inpatient ward grouped into four WASH domains, to be assessed and scored in agreement to written guidance (water [6 indicators]; sanitation [11 indicators]; hygiene [12 indicators]; organisational management [5 indicators]). An overall facility-level assessment guide at the end of the booklet was must also be completed, which contains 65 indicators also grouped into the 4 WASH domains (water [14 indicators]; sanitation [22 indicators]; hygiene [18 indicators]; organisational management [11 indicators]). The facility-level scoring entails a global assessment spanning all hospital areas, including outpatient services, kitchen, laundry, waste disposal infrastructure, the outdoor environment, and prior assessment of inpatient wards. Written guidance helps surveyors allocate a +, ++ or +++ score to each indicator, which corresponds to 0, 1 or 2 in the numeric scale used to calculate aggregate and percentage scores. The survey booklet provides text boxes to document field findings and explain the rationale for individual indicator scores. Indicator scores can also be used to assess specific WASH domains and sets of indicators linked to different levels of accountability. At ward level, 32 of the 34 ward indicators can be assigned for action/accountability. We use 16 indicators linked to the overall hospital management domain and 16 to the infection prevention and control committee (IPC) to explicate these uses [8].

A total of 19 health workers comprising doctors, nurses, pharmacists and public health officers, were recruited before the study to assist with the survey. The health workers were seconded to take part from the participating hospitals based on their familiarity with, and interest in improving infection prevention and control and WASH in their premises. Prospective surveyors were trained by research team supervisors in February 2018 for one week. The training comprised a theoretical and practical introduction to WASH, the WASH-FAST tool and the survey standard operating procedures and included participation in a one-day survey pilot at a district hospital with similar traits to facilities participating in the study. WASH-FIT training modules from the World Health Organisation (WHO) were adapted for the training of surveyors, while research team supervisors received WASH-FIT training directly from WHO.

Data collection commenced one week after the training and pilot and continued over two months (February- March 2018). Due to practical and logistical considerations, surveyors were divided into three teams, each composed of 4–5 surveyors plus one research team supervisor. Teams were then allocated a sample of hospitals for assessment at either western Kenya (five Hospitals), central Kenya (four hospitals) or around the capital city Nairobi (five Hospitals). At survey sites, the surveyor team was joined by 2–3 hospital representatives with specific roles as infection prevention and control coordinators or public health officials. This approach facilitated the training of local focal persons and built on-site capacity to undertake to follow up WASH assessments independently in future.

Data collection began with a meeting with the hospital management to collate information on the layout of the facility and retrieve the complete list of wards and service areas. This was followed by a walk-through of the hospital, noting the general external and indoor environments as well as any new or old buildings and infrastructure. A thorough assessment of each eligible ward was then conducted. The assessment included inpatient wards in the paediatric, medical, surgical and neonatal departments but excluded units not present in all hospitals (i.e. critical care, Ear Nose and Throat (ENT), eye, renal and psychiatric units). In each eligible ward, ward assessment forms were completed. Once these ward level inspections were

complete, there was an inspection of the entire facility, including the laundry, kitchen, outpatient areas and the external environment. Each indicator was assessed by direct observation and the score determined by team consensus on a three-point scale (meets = 2, partially meets = 1, or does not meet = 0 the required standard). A detailed explanation of the rating given was also provided in text notes by the data collection team. In each facility, data were collected over four days.

WASH is highly dependent on a range of health systems factors [10]. To understand the causes for the performance outcomes we measured, and the context of the survey results, the underlying health system components that support WASH activities were also investigated, including the availability of 'soft' and 'hard' infrastructure, material resources, local guidelines, and appropriate budgets. In addition, interviews were conducted with 17 hospital managers (e.g. medical directors, nursing and laboratory heads) and 14 frontline health workers (e.g. consultants, medical and nursing officers) in seven of the 14 hospitals. The sample was chosen to capture different socio-demographic influences on WASH and to determine the generalisability of the observations across the survey sites. To this end, interviewees also comprised a balanced mix of gender, age and experience. The design of the original semi-structured interview instrument was informed using informal discussions with stakeholders and experienced Kenyan clinicians within the research team. The instrument was subsequently revised organically throughout the research to reflect new insights garnered through earlier interviews. These interviews were conducted by the first, third and fifth authors. Each interview lasted 30–90 minutes. All the authors are well vast with the Kenyan context and have conducted research in Kenya. The first author is a Kenya doctor with vast experience working in the Kenyan health system and offered guidance to the interviewers during this process. All the interviewees were approached under the guidance of the hospital directors. Background information concerning the study and interview was availed to these interviewees before consenting which was done in writing. None of the respondents declined to give informed consent. These interviews were conducted in a quiet area of the hospital by one or two members of the study team and were audio-recorded.

There were no repeat interviews conducted during this study after reaching saturation. Although the transcripts were not returned to the respondents, general anonymised feedback of the study was provided to hospital management in each hospital.

All interview tools and information sheets are available on request. A (Consolidated criteria for Reporting Qualitative research) COREQ checklist was completed and is included as a supplement. (S2 File)

## Data quality assurance

During the study, data on WASH were entered into paper forms in the booklets provided. At the end of each day, the study coordinators together with the teams reviewed the data entered to ensure all the indicators were assessed and correctly documented. In case of any missing or unclear entries, the study team made specific reassessments the following day to obtain the data. This was done for all sites before moving to the next hospital. After completion of data collection, data from the paper forms were entered electronically into a database. To ensure data quality, double-entry was done and counterchecked by two members of the study team. All the interview recordings were stored securely before being transcribed verbatim.

## Data analysis

For analysis, to compute aggregate scores, the three indicator levels: Does not meet the target, partially meets and meets target were assigned numeric scores 0, 1, and 2 respectively and

individual indicator scores summed within the pre-specified levels of whole facility or ward or WASH domains or levels of responsibility. There were two main scoring approaches used for data analysis.[8]

1. **Aggregate percentage scores within hospitals generated at ward and facility level**
   Percentage scores were derived after summing the numerators and dividing these by the sum of denominators (representing the maximum possible score) for sets of individual indicators as described above. We used four categories to generate a colour chart with a "modified traffic light" system to display results using cutoffs of <40%-red, 41–60%-yellow, 61–80%-orange and 81–100%-green.

2. **Aggregate performance across hospitals at domain and indicator level**
   The overall domain percentage across the 14 hospitals was derived after summing the individual hospital indicator scores for each of the four domains (numerator) presented as a proportion of the maximum possible score. At the indicator level, the overall indicator percentage by domain was computed as the total score of each indicator from all the hospital wards as a proportion of the total maximum possible score.

These results are presented using simple histograms/bar charts to represent results and variations, and in the case of ward level percentage score, a scatter plot was used to describe each score and their variation and median score. All quantitative data analysis was conducted in R [11].

## Qualitative data

The audio files were transcribed and uploaded into Nvivo 12 [12], and the audio files were kept on an encrypted laptop. The third and fifth researchers coded the transcripts independently before discussing the codes and agreeing on combined axial codes. These discussions also involved the first author who used his knowledge of the survey data to make links between the survey findings and the open codes generated from the transcripts. We then arranged the axial codes according to the focus areas of the WASH-FAST survey,

## Results

### 1.WASH-FAST

In this section, we present the results of the WASH survey at the facility level and ward level. At ward level, these results are based on the specific WASH domains and by levels of responsibility. We then use data from the qualitative interviews to shed light on some of the quantitative findings from the WASH indicator survey. We excluded two hospitals from our analysis (H12 and H15). H12 is a rural health centre and therefore not comparable to the other more extensive facilities. In H15, which is a national referral hospital, we only assessed the neonatal unit and thus not able to compare with the other hospitals.

The bed capacity for the 14 participating hospitals ranged from 131 to 594 beds and the number of wards from 5 to 14. A total of 116 (85.3%) out of 136 wards were assessed in the survey (Table 1). Of the 20 excluded wards, six were renal; five were psychiatric, four were Ear Nose and Throat (ENT), three were Intensive Care (ICU) and one each was an eye and a neurosurgical unit.

**Performance-based on facility-level scores.** Based on the 65 indicators assessed spanning all four domains at the facility level, we present overall performance in each hospital in Fig 2 (vertical bars). Performance varied from 47% (H1) to 71% (H6) with a median of 61% IQR [56–65]. The central grid in Fig 2 represents the performance of the four domains in each

**Table 1. Summary of hospital size and wards assessed.**

| Facility | Hospital Bed capacity | Number of specialist doctors (consultants) | Number of wards in the facility | Wards assessed | Wards evaluated by Specialty | | | | | Wards Excluded |
|---|---|---|---|---|---|---|---|---|---|---|
| | | | | | Medical | Mixed Medical Surgical | Neonatal Unit | Paediatrics | Surgical | |
| **High Malaria Prevalence Zone** | | | | | | | | | | |
| **H1** | 203 | 5 | 7 | 6 | 3 | 0 | 0 | 1 | 2 | Renal Unit |
| H3 | 550 | 12 | 14 | 11 | 4 | 2 | 0 | 1 | 4 | Psychiatry, Renal Unit, ENT[a] |
| **H7** | 180 | 7 | 6 | 6 | 3 | 1 | 0 | 1 | 1 | None |
| H8 | 250 | 14 | 8 | 7 | 3 | 1 | 0 | 1 | 2 | Renal Unit |
| **H14** | 165 | 5 | 5 | 5 | 1 | 3 | 0 | 1 | 0 | None |
| **Low Malaria prevalence Zone** | | | | | | | | | | |
| **H2** | 594 | 26 | 16 | 12 | 5 | 1 | 1 | 1 | 4 | ENT[a], ICU[b], Psychiatry, Renal |
| H4 | 216 | 8 | 7 | 7 | 3 | 1 | 0 | 1 | 2 | None |
| **H5** | 231 | 7 | 9 | 8 | 3 | 1 | 1 | 1 | 2 | Psychiatry |
| H6 | 383 | 17 | 10 | 9 | 3 | 2 | 1 | 1 | 2 | Neurosurgery |
| **H9** | 550 | 19 | 18 | 14 | 4 | 2 | 1 | 2 | 5 | ENT[a], Renal, Psychiatry, ICU[b] |
| **H10** | 131 | 24 | 6 | 6 | 2 | 1 | 1 | 1 | 1 | None |
| **H11** | 320 | 21 | 9 | 9 | 4 | 3 | 1 | 1 | 0 | None |
| H13 | 378 | 20 | 15 | 10 | 4 | 1 | 1 | 1 | 3 | ENT[a], Renal, Psychiatry, ICU[b], Eye |
| **H16** | 350 | 14 | 6 | 6 | 2 | 1 | 1 | 0 | 2 | None |
| **Total** | | | **136** | **116** | **44** | **20** | **8** | **14** | **30** | **20** |

[a] ENT: Ear Nose and Throat

[b] ICU: Intensive Care Unit

hospital. Two hospitals (H4, H5) had an aggregate score of <40% in the hygiene domain (represented by red tiles in the colour chart). The domain scores for pooled hospital data are presented by the horizontal bars in Fig 2. From these horizontal bars, we note the hygiene domain performed poorest at 57%, and all the overall four domain scores are below 80%.

**Ward level performance.** The ward specific scores are represented by dots in the scatter plot in Fig 3. A median within hospital ward score (blue vertical lines in Fig 3) based on an all hospital's ward specific scores was generated for each of the WASH domains. These median domain scores were all less than 60% (water domain 59%, IQR[48–67], sanitation 55%, IQR [46–59], hygiene 54%, IQR[41–62] and organizational management 40%, IQR[30–60]). There was notable variability in the ward performance within hospitals, most marked in the water domain in hospital H9 in which 14 wards were assessed (range 40–90%). To compare this ward level performance with the overall score in the same facility, we include the overall facility level for each domain in each hospital (represented by non-shaded circles in Fig 3). By arraying the individual ward performance within each WASH domain for each hospital, we note clusters of underperformance mainly in the sanitation and management domains, where variability is substantial and outlier wards. The substantial differences between facility-level performance and ward performance are attributed to the inclusion of assessments spanning service areas in the overall facility score not captured in ward-based scores. We provided as an appendix the individual ward level performance (minimum and maximum score, mean, median and IQR) for each of the 34 indicators (S1 Table).

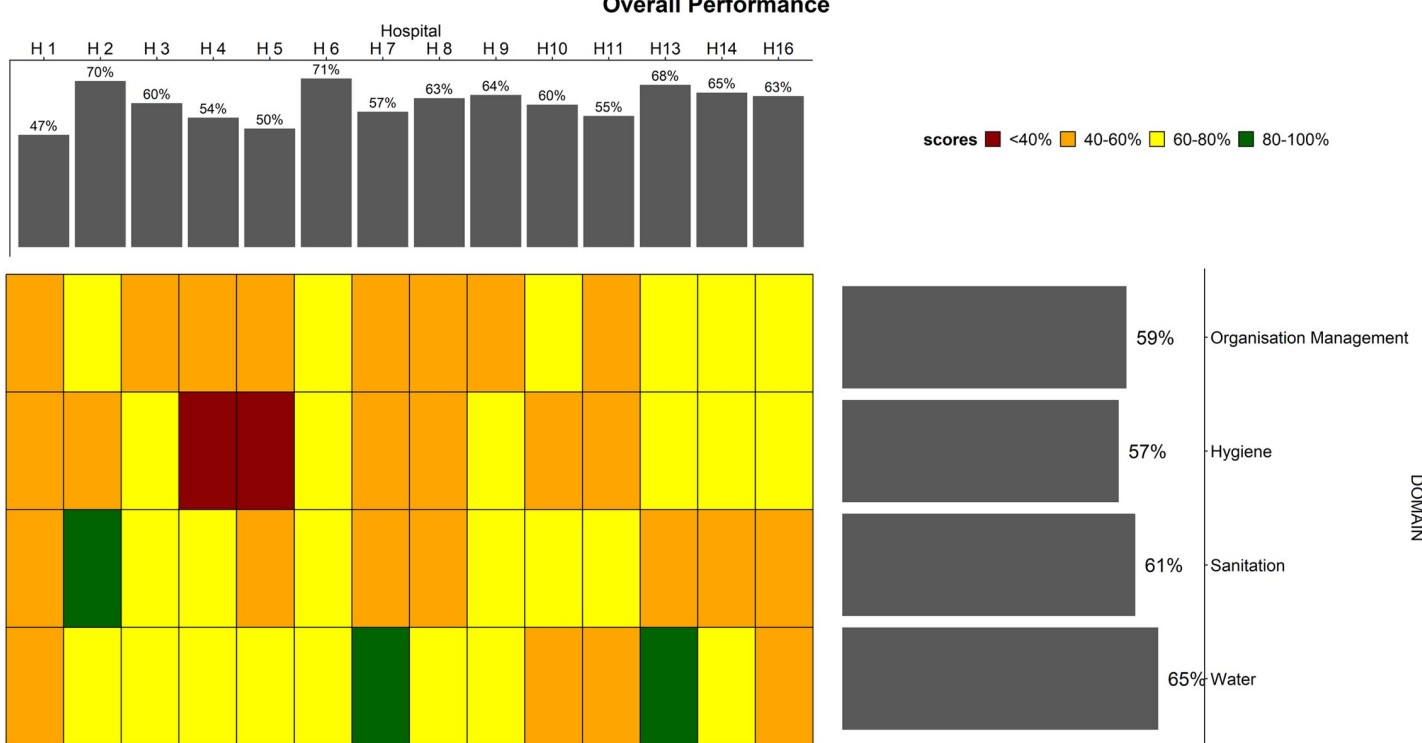

**Fig 2. Overall WASH performance.** The overall WASH facility performance based on all 65 indicators in four domains is shown by the upper vertical bars. The right horizontal bars summarise the performance of each domain across 14 hospitals. The tiles in the central grid are coloured according to the performance classification of each domain in each hospital, as shown in the colour legend.

**WASH domain performance.** Here, we examine how the individual indicators in the domains performed at the ward level. In Fig 4, for each specific indicator in the sanitation domain, we present the mean performance for each hospital (colour chart in the central grid). Also shown is the mean performance for each indicator across all the 116 wards (the horizontal bars in Fig 4) and the mean ward performance in the sanitation domain for each hospital (vertical bars in Fig 4). The WASH indicators that performed poorly in H11 (indicated by red tiles under H11 in the central colour grid) are waste management standard operating procedures, waste segregation, and availing toilet handwashing stations and cleaning records. We provide similarly constructed figures for the water, hygiene and organisational management domains in S3 File. Drawing from these additional figures, other poorly performing areas were drinking water storage (8%) and ensuring a minimum 2.5-metre distance between hospital beds in the hygiene domain (11%).

**Ward level performance by levels of responsibility.** Ward level data are also presented based on levels of responsibility. At ward level, we focus on two primary levels, indicators that need the action of senior hospital managers (16 indicators) and those reflecting activities the management can delegate to an infection prevention and control committee (16 indicators). The overall hospital performance for the indicators under the hospital management was between 44–65% with a median score of 56% IQR[36–57]. Of the 16 indicators, 10 had a score below 60% with poor performance noted on indicators that involved infrastructure (built environment). These include the availability of hand hygiene stations at points of care(53%) and service areas(51%), availability of connected taps(58%) and stations for drinking water(34%) and storage(8%). For the indicators under the IPC committee, the overall score ranged from

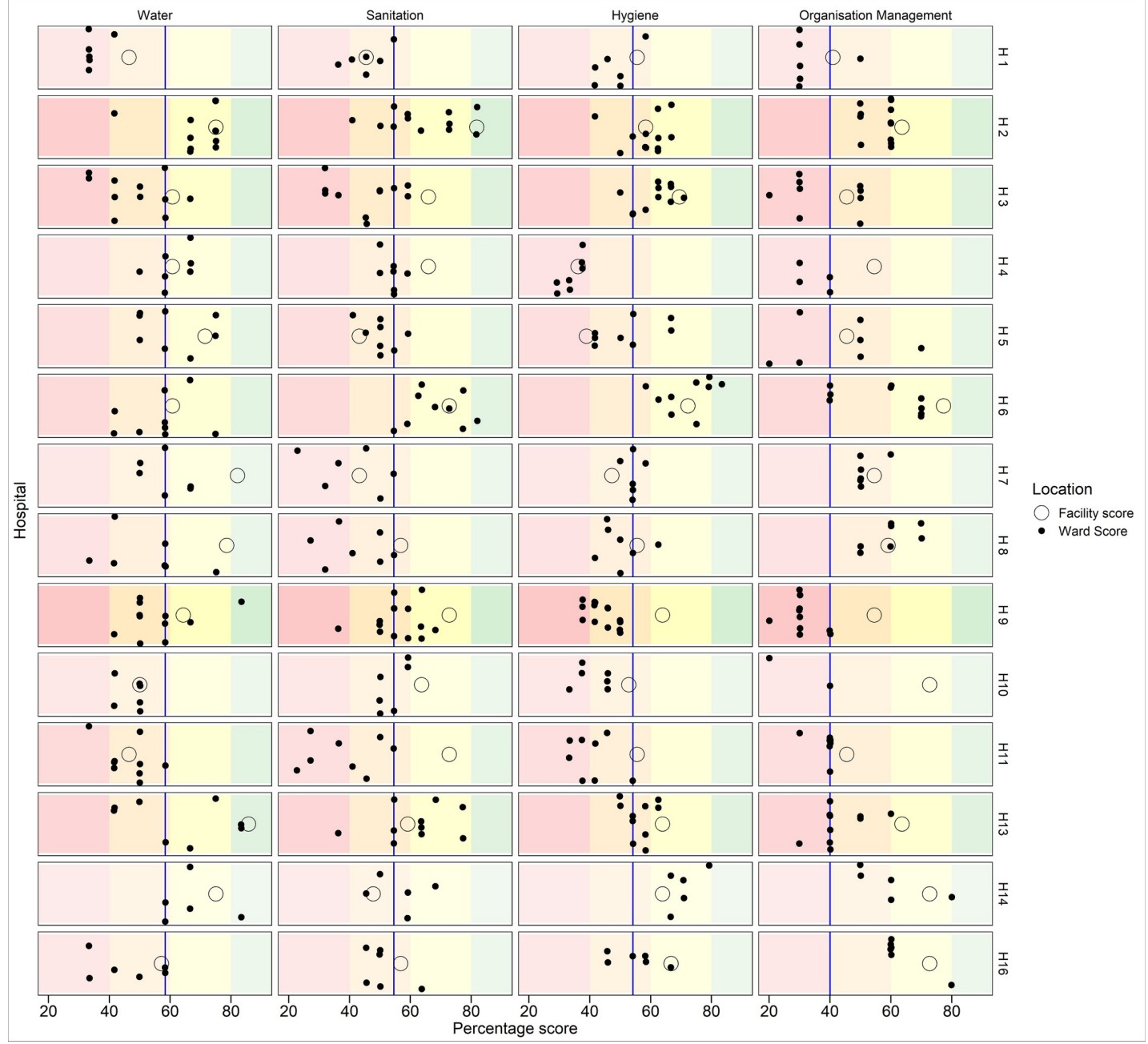

**Fig 3. Ward level WASH performance.** Horizontal scatter plot of the aggregate ward level scores of 116 wards (black shaded points) across 14 hospitals by domain. Also included is overall facility aggregate score(O) for each hospital by domain. The overall facility score includes assessment of inpatient wards and other service areas across the hospital. The blue vertical line in each domain represents the median ward score for that domain. The colour bars represent cut off values of red <40%, orange41-60%, yellow 61–80% and green 81–100%.

30–70% with a median score of 45% IQR[36–57]. There were 11 indicators with a score of less than 60%. The figures describing these indicators are provided as supplement (S4 File)

## 2.Understanding variation–qualitative analysis

**1. Challenges with the built environment.** Many of the facilities we surveyed are more than 40 years old and have not been renovated or modernised, or if they have been, this has

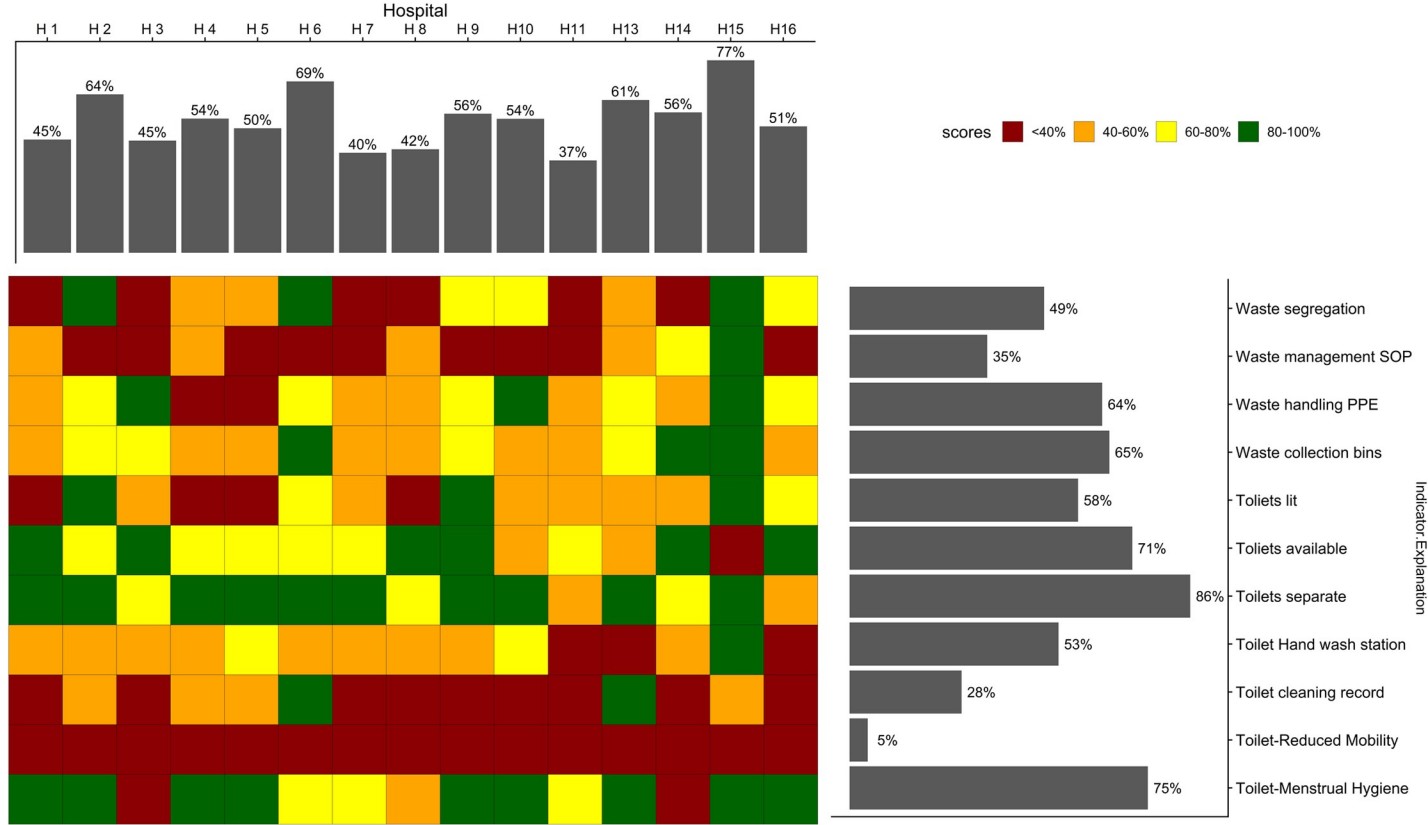

**Fig 4. WASH performance for sanitation domain at ward level.** Shows overall service performance at the ward level for sanitation domain with 11 indicators shown by the vertical bars. The horizontal bars summarise the performance of each indicator across all the hospital wards. The tiles in the central grid are coloured according to the performance classification of each indicator in each hospital as per the colour legend. SOP: standard operating procedures, PPE: personal protective equipment.

been done in a partial way that affects only individual wards or areas of the hospital. This leads to variation both between and within facilities. The newer hospitals and ward units had more sinks, and toilets available, and their arrangement into cubicles compared to an open floor plan improved bed spacing. The workers seemed to take such arrangements as fixed and beyond their control, but health workers recognised the importance of the built environment:

> "*The sinks like the ward I am in, and I have seen the other ones as well, there are sinks in every cube, that is about eight beds, there is a sink in every eight bed a sink.*"

(Health worker)

> "*Right now, I talk of where I am working in the female ward and the dressing room we are missing the sink between the beds. We need a sink for cleaning hands in that place. Right now, there is no sink.*"

(Health worker)

The built environment and the quality of the facilities enable or retard progress in WASH. For both facilities that scored red in 'Hygiene' (Fig 1), the facilities were outdated and difficult to keep clean. Not having sinks available makes life difficult for these facilities and the results of the survey were reflected in respondents' comments on potential underperformance:

"*But how we do it in [H5] we don't really have, you know, a tap everywhere and soap everywhere.*"

(Nurse)

In addition to the overall condition of some wards, the design and layout of hospitals also seemed to play a part. Some of the buildings in these hospitals were built more than 40 years ago at a time when essential IPC and WASH standards were perhaps not prioritised and were constructed with small populations in mind. Therefore, the regional governments and hospital managers have inherited hospitals whose size and structure are indeed not within their control. This was described by one of the regional government leaders who highlights challenges with space for waste collection.

"*Another challenge I want to say is space because you will find that in a good set up, we need to have a space for collecting the waste that is specifically labelled waste collection area. Whereby the officers who transport the waste from the generating area to the disposal area, should have a designated area which is not in our hospital at the moment, we are just improvising. But in most places, that are well constructed, they are places you will even lock so that the issue of contamination is minimal. In this county except for the new lab that has a designated waste collection area.*"

(Regional government officer)

**2. Resource availability and allocation.** The budgets available for IPC and WASH very much varied by region. Some of the facilities in the survey reported challenges with budgetary allocation at the county government level for provision of WASH materials leading to shortages in supply or having sub-standard materials, as highlighted by one of the IPC hospital leaders whose facility had to contend with ill-fitting personal protective equipment.

"*The challenge is inadequate funding now that you are given some money that cannot meet your expectations then that one comes as a challenge because perhaps you want to buy a certain amount of material, but because of limited funding you buy as per what is available.*"

(Health Manager)

Importantly, IPC rarely had a specific budget, and our respondents explained how this made it challenging to ensure supplies:

"*I would talk about maybe the sanitary requirement that we need to help us do prevention but not IPC budget as per se. There is nothing like that.*"

(Frontline healthcare worker)

This is important because it results in splitting the responsibilities for ensuring regular supplies across several areas and individuals:

"*You know normally we are told to itemise whatever we require in the departments that we are working. Like now the public health officer is the one who deals with the sanitation and such issues. [But] chlorine it is under the pharmacy, it is under non-pharmaceutical commodities, so we give her, the pharmacist a figure or the requirements of the consumption in the institution and he or she factors that.*"

(Frontline healthcare worker)

This may also speak to some of the variations we see between wards in a single facility where some individual ward managers are better at procuring funds for IPC materials than others (see Fig 3).

**3. Leadership at hospital and ward level.**   In some of the facilities, hospital managers did not appear to see WASH/IPC as a priority area, and there was no institutional leadership or commitment to tackle WASH challenges.

> "*you know change of the management brings a tug of war, and people normally have their own ..there is what they value. What I want basically to tell you people have not taken infection prevention as a key concept in health sector they take lightly . . ..when you come to certain managers, there are those who don't value infection prevention they see it as by the way.*"

> (IPC Lead)

In facilities that focused more on WASH, attempts were sometimes made to get better representation from the ward level leaders, such as hospital consultants and senior nurses, to take leadership roles in IPC committees to mobilise broader support for WASH. These doctors and senior nurses seemed vital in improving WASH at the ward level.

> "*like now . . . we have a paediatrician who is now chair- chairing our committee for infection prevention, but he's working in the paediatric unit. . . That is now what makes it easier because now for the doctors. . . if there is something that now the doctors needed to be communicated to, we use him.*"

> (Ward Manager)

Local leadership can also affect hospitals and result in variable performance between wards, as seen in Fig 3. These wards were often those where patients were understood to be vulnerable such as the NBU and had persons who enforced improved IPC behaviour:

> "*I think it is because we have somebody who is very vocal. And she is very strict. She will tell them you are not going to handle any baby without having washed your hands.*"

> (Health Manager)

A manager in one of the better-performing facilities, noted the positive effects of having such a "champion" at the facility level, leading the overall approach and IPC committee:

> "*I think the nurse the in charge–the unit matron who is in charge of that department [IPC/WASH]–is somebody very passionate about the department, and more so you'll find even she keeps on pushing us.*"

> (Hospital Deputy director)

### 3.Improving wash–ideas from the field

In addition to the critical areas of facilities, budget and leadership, we also identified other key insights from the qualitative interviews that affect WASH performance at ward and facility levels. These would be key areas to consider for intervention to improve WASH and IPC in hospitals

**1. Outsourcing–a solution for general cleanliness?.**   With the approval of the county governments, to combat challenges of staff shortages, supplies and poor accountability, some of

the hospitals had begun to outsource cleaning services. These companies, at a fixed price contract, provided personnel and supplies and were responsible for ensuring these facilities were clean at all times.

> "*the cleanliness in this place in comparison with most of the other you know public institutions in this country is different more so the people you see sweeping; cleaning are not employees of the ministry they're not employees of the hospital we have outsourced it to somebody who has been doing it in other various places.*"

(Deputy Hospital Director)

This strategy did not seem to work in all facilities as some of the poorly performing hospitals like H11 also had outsourced cleaning services. It may still come down to the hospital management to ensure these companies are working well.

**2. Improving personal and professional attitudes towards WASH and IPC.** Compared to other areas in the hospital, WASH and IPC are not taken as seriously, mainly because these are not revenue-generating activities. These have also been viewed historically as the premise of some particular cadres like nurses. Therefore, doctors and other medical specialists have found it difficult to take orders from WASH focal persons who may be junior to them.

> "*. . .sometimes when you go for this IPC meetings, and something on IPC comes up you could feel some staff complaining, especially the nurses that doctors don't embrace the issues of IPC. The other issue also is the attitude of people; some people have very bad attitudes. They imagine I'm being . . . it's like you are bothering them.*"

(Nurse Manager)

The nurses we spoke to also noted that it was sometimes hard to persuade other cadres of the importance of hand washing:

> "*I think they[Nurses]are more conversant with the infection prevention issues, and they observe the protocols but the other cadres, they don't observe. I think they are not. Maybe they are not updated as to why they should observe the protocols and they, by the way, I would say, they don't mind about the way they do their things.*"

(Nurse manager)

In contrast, other hospitals, the perspective is different and more positive. Some health workers see WASH and IPC as key to the provision of care in the hospital.

> "*IPC is very important because I think to me it's the heartbeat of the hospital.*"

(Frontline Health worker)

**3. Training and orientation of all cadres of staff on WASH.** To correct some of the poor attitudes and challenges of WASH and IPC. Some hospitals have taken up the training of all cadres to WASH and IPC as part of the hospital orientation to ensure everyone is up to date.

> "*its routine that like now when you get posting all people nurses and clinical officers and medical officers' interns when they come here now it has become even before they get into the ward*

*that's [orientation on IPC]the first place they step in then they are taken through and it has become very much entrenched . . . in their absorption into this place"*

(Deputy Medical Director)

**4. Partnerships to improve WASH offer partial improvement in the sector.**  Some hospitals, due to constraints in funding, have linked with development partners to provide training (e.g. H10 and H16), assist in the development of standard operating procedures and promotional materials. Although these partners have a positive effect, we noted in some hospitals, duplication of roles, for example, multiple different partners engaged in promoting hand washing.

Like for the [partner X] they had the concept, they taught us about their hand washing, the decontamination, the waste segregation the storage and about the how to care for the laundries and all that. All those steps were taught but [partner Y] they came in the same, but they narrow to the hand washing.

(Hospital manager)

## Discussion

Here we report the combined hospital performance of the four WASH domains in 14 hospitals. Performance varied at hospital, ward and indicator level with most being sub-optimal. Some of the themes explaining this variation and performance included differences and challenges within the built environment, resource allocation, leadership, training on WASH and health worker attitudes towards WASH and IPC.

The JMP also assessed some of the indicators assessed in WASH-FAST in the 2016 global assessment of WASH in health facilities [4]. Although using different methodologies, our results confirm and highlight similar issues. The main areas showing the same poor performance with the JMP are the WASH infrastructure, including water and toilet availability and health care waste management. The JMP and other related WASH surveys have shown slow progress in the improvement of WASH in most hospitals in resource-limited settings. This is despite improving WASH in health care facilities being highlighted as one of the sustainable development goals [13].

WASH in health facilities is a crucial intersection point for many hospital-based interventions. These include IPC, control of antimicrobial resistance, improving the safety and quality of care and health system strengthening interventions all aimed towards improving efficiency and patient outcomes in hospitals. A focus on improving WASH would, therefore, have a positive effect on multiple aspects of health care. Some of the main inputs needed to improve WASH in health facilities are political resources, financial/material resources and human resources [2]. The leading players to provide these inputs are the national and regional governments, policymakers and at hospital level the senior hospital managers who may also delegate to specific committees.

Political resources include engagement from the national and county governments and development partners. These provide national standards/guidelines and accountability mechanisms to ensure that facilities are meeting these national standards [14]. From our survey, the results and findings that that would be relevant to national/regional governments are the findings at the overall facility level. These show varied hospital performance across the country. The role played by the government, in this case, would be to ensure all hospital achieve a

minimum standard for WASH. For example, a directive from the national government on standards for building hospitals would provide minimum standards for WASH infrastructure are present in every facility especially for newer hospitals and in older hospitals that may be undergoing renovation. These governments are also crucial in positioning WASH on the national health agenda linked to the sustainable development goals, universal health coverage [2] and the prevention of AMR to help make WASH a priority for all.

Inadequate resource allocation was highlighted as a challenge affecting the performance of WASH in these facilities. To improve WASH, there needs to be resource allocation from the national/county government with budgets for policy generation, assessments, upgrading of infrastructure and training for WASH. At the hospital level, to improve and maintain the WASH infrastructure and practices, resources are required for ongoing staff training supported by an IPC/AMR committee, repairs need to be prioritised, and a constant supply of quality materials should be ensured with regular assessments of all the wards performance. Resources for WASH assessments and rewards to staff for best performing wards may also need to be allocated [14, 15]. In some of the hospitals, some challenges were overcome by partnering with development agencies to provide extra resources to improve WASH in the hospitals. Hospital and ward level WASH data as provided in this survey would be essential to highlight the main areas or indicators within the hospital or wards that need to be prioritised during resource allocation. We noted at ward level in these hospitals that there was poor performance for some critical indicators that would be quick and less costly to improve. These include; availing a cleaning record in the wards (15%) and toilets (27%) and providing waste management standard operating procedures (SOP) (30%) and hand hygiene promotion materials(42%).

Human resource allocation within hospitals is also crucial to improving WASH. When these staff, including cleaners and waste handling personnel, are well trained and motivated, they will provide better service. All the workers, whether employed or contracted need to be adequately trained and oriented on WASH and IPC. In some of the hospitals we assessed, this process of training and orientation was now routine. Training on WASH can be formal preservice or in-service training and mentoring. To achieve these levels of training and competence in these hospitals, there is a need for some trained WASH focal persons to provide technical expertise in hospitals [2]. These focal persons might be recognised through certification, as in the case in some African countries [6]. In facilities with staff shortages, outsourcing of some of these workers from cleaning companies could improve efficiency, help hospitals focus on their core business of providing quality healthcare and may contain costs [16]. The hospital managers are, however, ultimately responsible for the performance of these outsourced services and therefore need to monitor the services provided actively, tools such as WASH-FAST could help this process.

This survey also highlights the role of committed leadership in improving WASH. Leadership at all levels of the health systems is an essential contributor to improving WASH and averting AMR. National leadership provides a clear road map for the country to follow and aligns all WASH/AMR efforts to achieve the set targets. It should offer tools and approaches for assessment of WASH/AMR in the country, which should be prioritised over those provided by other donor partners to ensure coherence [14]. Leadership at the hospital level is also critical in improving WASH/AMR. Hospital managers are vital to improving staff morale and attitudes to embrace WASH as part of their core functions, for example, by including these functions in the staff appraisal. Hospital management can also delegate some of the WASH activities to committees within the hospital. In this case, a well-constituted IPC committee would be essential in offering training on WASH, IPC and AMR, and conducting periodic assessments of the hospital and offering technical and budgetary recommendations to the

senior hospital managers[8]. Including senior doctors and nurses who actively participate in these committees can enhance their and effectiveness credibility and improve attitudes clinicians have towards WASH[17]. Having champions of change (WASH champions) at the ward level for activities like hand washing has previously been noted to generate positive change [15]. This approach of using champions may also prove valuable in improving the overall outlook of WASH in healthcare facilities, as shown in some of the hospitals surveyed.

Financing for health in many countries, including Kenya remains a challenge. The current budget for health below 8% of the total national budget and this is still below the global recommendations.[18]. This proportion has not increased for several years despite new challenges like AMR emerging in recent years. In Kenya with a devolved health system, the amount of money that would be set aside for activities like buying IPC and WASH materials and other non-pharmaceutical agents was only 5% of the budget with more than 70% of the county budgets for health going to personnel costs.[18] Therefore, to improve WASH, IPC and the AMR agenda, especially in public hospitals, there is a need to examine other sources of funding for these activities. Public-private partnerships, donor agencies may provide alternative ways to fund these activities in limited-resource settings. In Kenya, the National hospital insurance fund, which is the national health insurance provides reimbursement to public and private facilities for some of the costs incurred by some of its members. Pegging some of these reimbursements to the quality of care provided, including the state of IPC and WASH in these hospitals may force these facilities to improve[19].

From our work using the WASH-FAST tools, we demonstrate that; surveys can be carried out at national level or regional levels to generate data on the state of WASH in healthcare facilities in countries. These data have the potential for use in priority setting for WASH interventions, policy generation and resource allocation both at the national and hospital level. These data generated at the hospital level can also be used by authorities for hospital accreditation and benchmarking.

The main limitations of our survey were that the WASH-FAST assessment tool provided for only three possible outcomes for each indicator. This may have implications in the accuracy/objectivity of the score presented, to mitigate this, we ensured all the clerks were well trained before the survey, and the data collection team provided notes against every indicator to ensure the score provided was as objective as possible. The other limitation in our approach was conducting the surveys and interviews over almost the same period. Although interviews were conducted after the survey teams shared their daily findings with the social scientists and some of the social scientists were also present during the surveys, a more sequential approach would have been preferable. That is, after completing and analysing the quantitative data, more relevant interview questions might have been formulated, tested, and interviews might then have been conducted across most survey hospitals. Recruiting health workers from participating hospitals to collect data was likely to introduce bias. To mitigate, we ensured that these data collection teams were a mix of health workers from the participating hospital and others from different hospitals and included a study team leader. The indicator scores assigned were also arrived at by team consensus. The main aim of including health workers from participating hospitals was to build capacity for these facilities to carry out similar surveys in future as part of their facility improvement strategies.

Future work using the WASH-FAST might explore how we could systematise feedback to hospitals of survey results and recommended follow-up actions, including the use of electronic dashboards. Currently, all the indicators have equal weight, but we acknowledge some may be more critical for patient safety or in the prevention of acquired infections. Future work would, therefore, consider ways to give more weight to more critical indicators during surveys, analysis, reporting and follow-up.

## Conclusion

Significant differences and challenges exist in the state of WASH within and across even large hospitals providing multi-speciality care. Whereas the senior hospital management can make some improvements, input and support from the national and county governments are essential to improve WASH as a necessary foundation for safe, quality hospital care and to avert AMR in Kenya.

## Supporting information

**S1 Table. WASH aggregate ward indicator scores.**
(DOCX)

**S1 File. WASH data collection and standard operating procedures tool.**
(DOCX)

**S2 File. Consolidated criteria for reporting qualitative studies (COREQ): 32-item checklist.**
(DOCX)

**S3 File. Ward level aggregate by domain for water, hygiene and organisation management.**
(DOCX)

**S4 File. Performance of indicators under the infection prevention and control committee and hospital management.**
(DOCX)

## Acknowledgments

The authors thank the Kenyan Ministry of Health and the Council of Governors who permitted this work to be carried out. The authors thank Dr Nancy Abuya and all the clinicians who assisted in data collection. We also thank the hospital management and clinical teams who supported the work in the survey hospitals. The authors also thank Arabella Hayter and Margaret Montgomery from the Water Sanitation, Hygiene and Health Unit, Department of Public Health and the Environment, WHO for providing WASH training materials and technical advice. This work is published with the permission of the Director of KEMRI.

## Author Contributions

**Conceptualization:** Michuki Maina, Olga Tosas-Auguet, Mike English.

**Data curation:** Michuki Maina, Mathias Zosi.

**Formal analysis:** Michuki Maina, Olga Tosas-Auguet, Jacob McKnight, Grace Kimemia, Paul Mwaniki.

**Funding acquisition:** Olga Tosas-Auguet, Mike English.

**Investigation:** Michuki Maina, Olga Tosas-Auguet, Jacob McKnight, Mathias Zosi, Grace Kimemia.

**Methodology:** Michuki Maina, Olga Tosas-Auguet, Jacob McKnight, Paul Mwaniki, Constance Schultsz.

**Project administration:** Michuki Maina, Olga Tosas-Auguet, Mathias Zosi, Mike English.

**Software:** Michuki Maina, Grace Kimemia, Paul Mwaniki.

**Supervision:** Michuki Maina, Olga Tosas-Auguet, Jacob McKnight, Mathias Zosi, Constance Schultsz, Mike English.

**Validation:** Michuki Maina, Olga Tosas-Auguet, Jacob McKnight, Grace Kimemia.

**Visualization:** Michuki Maina, Olga Tosas-Auguet, Jacob McKnight, Paul Mwaniki.

**Writing – original draft:** Michuki Maina, Jacob McKnight.

**Writing – review & editing:** Michuki Maina, Olga Tosas-Auguet, Jacob McKnight, Mathias Zosi, Grace Kimemia, Paul Mwaniki, Constance Schultsz, Mike English.

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
