## [Decision Letter · Decision Letter 0]

16 Aug 2019

PONE-D-19-19421

EVALUATING THE FOUNDATIONS THAT HELP AVERT ANTIMICROBIAL RESISTANCE: PERFORMANCE OF ESSENTIAL WATER SANITATION AND HYGIENE FUNCTIONS IN HOSPITALS AND REQUIREMENTS FOR ACTION IN KENYA

PLOS ONE

Dear Dr Maina,

Thank you for submitting your manuscript to PLOS ONE. After careful consideration, we feel that it has merit but does not fully meet PLOS ONE’s publication criteria as it currently stands. Therefore, we invite you to submit a revised version of the manuscript that addresses the points raised during the review process.

We would appreciate receiving your revised manuscript by Sep 30 2019 11:59PM. To enhance the reproducibility of your results, we recommend that if applicable you deposit your laboratory protocols in protocols.io, where a protocol can be assigned its own identifier (DOI) such that it can be cited independently in the future. For instructions see: http://journals.plos.org/plosone/s/submission-guidelines#loc-laboratory-protocols

We look forward to receiving your revised manuscript.

Kind regards,

Lars-Peter Kamolz, M.D., Ph.D., M.Sc.

Academic Editor

PLOS ONE

Journal Requirements:

a) Please provide an amended Funding Statement that declares *all* the funding or sources of support received during this specific study (whether external or internal to your organization) as detailed online in our guide for authors at http://journals.plos.org/plosone/s/submit-now.  

b) Please state what role the funders took in the study.  If any authors received a salary from any of your funders, please state which authors and which funder. If the funders had no role, please state: "The funders had no role in study design, data collection and analysis, decision to publish, or preparation of the manuscript."

Reviewers' comments:

Reviewer's Responses to Questions

**Comments to the Author**

1. Is the manuscript technically sound, and do the data support the conclusions?

Reviewer #1: Yes

Reviewer #2: Yes

2. Has the statistical analysis been performed appropriately and rigorously? 

Reviewer #1: Yes

Reviewer #2: Yes

3. Have the authors made all data underlying the findings in their manuscript fully available?

Reviewer #1: Yes

Reviewer #2: Yes

4. Is the manuscript presented in an intelligible fashion and written in standard English?

Reviewer #1: Yes

Reviewer #2: Yes

5. Review Comments to the Author

Reviewer #1: In my opinion the manuscript is well-written with all necessary details, quite informative and important in the field. It was a nice read for me and I have no objection against it to be published in PLOS ONE.

Reviewer #2: It appears that the participating health workers were recruited from the participating hospitals what I think could have caused bias. If I misunderstood this, please clarify.

6. PLOS authors have the option to publish the peer review history of their article (what does this mean?). If published, this will include your full peer review and any attached files.

Reviewer #1: No

Reviewer #2: No

---

## [Author Response · Author response to Decision Letter 0]

29 Aug 2019

Reviewer Comments

It appears that the participating health workers were recruited from the participating hospitals what I think could have caused bias. If I misunderstood this, please clarify.

Response 

We thank the reviewers for taking time to review this manuscript. In response to the above comment;

 We acknowledge that using health workers from participating hospitals to collect data might introduce bias. To mitigate this, we ensured that these data collection teams were a mix of health workers from that particular hospital where the data collection was taking place and others from the other participating hospitals. This team also included a study coordinator. The indicator scores assigned and documented were all arrived at by team consensus. The main aim of including health workers from participating hospitals was to build capacity for these facilities to carry out similar surveys in future as part of their facility improvement strategies. We have included this information in the manuscript (Page 26 Line 551-557)

---

## [Decision Letter · Decision Letter 1]

11 Sep 2019

[EXSCINDED]

Evaluating the foundations that help avert antimicrobial resistance: Performance of essential water sanitation and hygiene functions in hospitals and requirements for action in Kenya

PONE-D-19-19421R1

Dear Dr. Maina,

We are pleased to inform you that your manuscript has been judged scientifically suitable for publication and will be formally accepted for publication once it complies with all outstanding technical requirements.

With kind regards,

Lars-Peter Kamolz, M.D., Ph.D., M.Sc.

Academic Editor

PLOS ONE

Additional Editor Comments (optional):

Reviewers' comments:

Reviewer's Responses to Questions

**Comments to the Author**

1. If the authors have adequately addressed your comments raised in a previous round of review and you feel that this manuscript is now acceptable for publication, you may indicate that here to bypass the “Comments to the Author” section, enter your conflict of interest statement in the “Confidential to Editor” section, and submit your "Accept" recommendation.

Reviewer #1: All comments have been addressed

Reviewer #2: All comments have been addressed

2. Is the manuscript technically sound, and do the data support the conclusions?

Reviewer #1: Yes

Reviewer #2: Yes

3. Has the statistical analysis been performed appropriately and rigorously? 

Reviewer #1: Yes

Reviewer #2: N/A

4. Have the authors made all data underlying the findings in their manuscript fully available?

Reviewer #1: Yes

Reviewer #2: Yes

5. Is the manuscript presented in an intelligible fashion and written in standard English?

Reviewer #1: Yes

Reviewer #2: Yes

6. Review Comments to the Author

Reviewer #1: After reading the revised manuscript, I have no objections for acceptance of the present manuscript for publication in PLoS ONE journal.

Reviewer #2: The authors have argued reasonably to my initial concern and I can now recommend to publish this paper.

7. PLOS authors have the option to publish the peer review history of their article (what does this mean?). If published, this will include your full peer review and any attached files.

Reviewer #1: No

Reviewer #2: No

---

## [Editor Report · Acceptance letter]

25 Sep 2019

PONE-D-19-19421R1 

Evaluating the foundations that help avert antimicrobial resistance: Performance of essential water sanitation and hygiene functions in hospitals and requirements for action in Kenya 

Dear Dr. Maina:

I am pleased to inform you that your manuscript has been deemed suitable for publication in PLOS ONE. Congratulations! Your manuscript is now with our production department. 

With kind regards,

on behalf of

Dr. Lars-Peter Kamolz 

Academic Editor

PLOS ONE